# Maternal Supplementation with Ornithine Promotes Placental Angiogenesis and Improves Intestinal Development of Suckling Piglets

**DOI:** 10.3390/ani14050689

**Published:** 2024-02-22

**Authors:** Yun Yang, Guanyu Hou, Fengjie Ji, Hanlin Zhou, Renlong Lv, Chengjun Hu

**Affiliations:** 1Tropical Crop Genetic Resource Research Institute, Chinese Academy of Tropical Agricultural Sciences, Haikou 571101, China; yangyun2023@126.com (Y.Y.); guanyuhou@126.com (G.H.); fengjie_ji@126.com (F.J.); zhouhanlin8@163.com (H.Z.); 2College of Animal Science and Technology, Huazhong Agricultural University, Wuhan 430070, China

**Keywords:** angiogenesis, placenta, ornithine, intrauterine growth restriction, intestinal development

## Abstract

**Simple Summary:**

This study aims to investigate the effects of maternal dietary supplementation with ornithine on productive performance and placental angiogenesis of sows. We found that maternal dietary supplementation with 0.15% ornithine increased the average birth weight of piglets and intestinal development of suckling piglets. In addition, Orn facilitated placental angiogenesis by regulating vascular endothelial growth factor-A (VEGF-A). These results indicated that dietary supplementation with 0.15% ornithine is an effective strategy to improve the productive performance of sows.

**Abstract:**

The blood vessels of the placenta are crucial for fetal growth. Here, lower vessel density and ornithine (Orn) content were observed in placentae for low-birth-weight fetuses versus normal-birth-weight fetuses at day 75 of gestation. Furthermore, the Orn content in placentae decreased from day 75 to 110 of gestation. To investigate the role of Orn in placental angiogenesis, 48 gilts (Bama pig) were allocated into four groups. The gilts in the control group were fed a basal diet (CON group), while those in the experimental groups were fed a basal diet supplemented with 0.05% Orn (0.05% Orn group), 0.10% Orn (0.10% Orn group), and 0.15% Orn (0.15% Orn group), respectively. The results showed that 0.15% Orn and 0.10% Orn groups exhibited increased birth weight of piglets compared with the CON group. Moreover, the 0.15% Orn group was higher than the CON group in the blood vessel densities of placenta. Mechanistically, Orn facilitated placental angiogenesis by regulating vascular endothelial growth factor-A (VEGF-A). Furthermore, maternal supplementation with 0.15% Orn during gestation increased the jejunal and ileal villi height and the concentrations of colonic propionate and butyrate in suckling piglets. Collectively, these results showed that maternal supplementation with Orn promotes placental angiogenesis and improves intestinal development of suckling piglets.

## 1. Introduction

Intrauterine growth restriction (IUGR) is characterized by the failure of fetuses to attain their inherent growth potential [1], and is associated with augmented risks of diseases, morbidity, and mortality of fetuses [2]. The blood vessels in the placenta play an important role in the exchange of oxygen/nutrients between maternal and fetal circulations [3]; their aberrant formation can curtail the supply of oxygen and nutrients to fetuses [4], thereby limiting fetal growth [5]. Consequently, understanding the mechanism of blood vessels formation in placenta can facilitate the development of therapeutic strategies for IUGR.

Ornithine (Orn), synthesized through the hydrolysis of arginine in urea cycle [6], is a key substrate for the synthesis of proline and polyamines [7]. Proline positively influences reproductive processes. For instance, maternal proline supplementation during gestation has been shown to augment the embryonic survival and placental development in mice [8]; short-term supplementation with proline could increase the litter size and birth-weight of piglets in sows [9]. Furthermore, a previous study indicated that proline modulates the angiogenesis of placentas by synthesizing polyamines [10]. Polyamines are essential for cell growth, performing critical functions in biological processes including protein synthesis and angiogenesis [10,11]. Ornithine decarboxylase (ODC), an enzyme catalyzing the decarboxylation of Orn to polyamines, was reported to facilitate endothelial cell angiogenesis in vitro [12]. These results suggested that maternal Orn supplementation might bolster placental angiogenesis via synthesizing proline and polyamines.

The intestine development during pregnancy is crucial for growth and health of piglets after birth. IUGR could lead to impaired gut morphology and function in piglets [13]. However, the small intestine of fetuses is affected by the maternal nutrient intake [14]. Previous study showed that maternal dietary supplementation with proline improves fetal pig growth and intestinal epithelial cell proliferation [15]; spermidine reduces intestinal inflammation and maintains a healthy microbiome [16], suggesting maternal nutrition intervention is an effective strategy for improving fetal intestinal development. However, the role of maternal Orn supplementation on intestinal development of piglets is unknown.

Therefore, the objective of the present study is to explore the role of Orn in placental angiogenesis utilizing a pig model and to discern whether maternal Orn supplementation influences the intestinal development of suckling piglets. Our findings provide insight into the role of Orn in placental angiogenesis and fetal growth.

## 2. Materials and Methods

### 2.1. Animals and Experimental Treatments

Experiment 1: A total of 8 primiparous gilts (Bama pig) with average body weight of 61.5 ± 0.76 kg was obtained from a pig farm of Hainan Guangfu Agricultural Development Co., Ltd., located in Baisha, Hainan province, China. The gilts were artificially inseminated with pooled semen from a Duroc boar and randomly assigned to one of eight pens, with one gilt per pen. The animals were fed a diet formulated according to the Chinese National Feeding Standard (2020) [17] for gestating gilts (Table 1). Each pen was equipped with a feeder and a nipple drinker. All gilts had *ad libitum* access to drinking water and were fed three times daily. Feeding schedules were partitioned as follows: 1.0, 1.5, 1.8, and 2.0 kg of diet per day during the gestational periods of days 1–30, 30–75, 75–90, and 91–110, respectively. To investigate the placental angiogenesis of low weight fetuses during pregnancy, gilts were euthanized at days 75 (*n* = 4) and 110 (*n* = 4) of gestation, respectively. Fetal and placenta weights were recorded. Two grams of each placenta (3 to 4 cm from the cord insertion point) was immediately snap-frozen in liquid nitrogen or immediately fixed in 4% paraformaldehyde. Subsequently, the fetal pigs with body weight close to average body weight were divided into normal weight group (N), and those body weight less than two standard deviations were divided to low weight group (L).

Experiment 2: A total of 48 gilts (Bama pig) were artificially inseminated with pooled semen from a Duroc boar and randomly assigned to one of four groups (*n* = 12/group). Gilts in the control group were given a standard diet (CON group), whereas those in the experimental group were fed the standard diet supplementation with 0.05% (0.05% Orn group), 0.10% (0.10% Orn group), and 0.15% Orn (0.15% Orn group), respectively. The nutrient level for the basal diet (Table 1) met the requirement of the Chinese National Feeding Standard (2020) for gestating gilts. Feeding schedules mirrored those previously described, including 2.50 kg of lactation diet (Table 1) per day during lactation. The gilts had free access to drinking water. Gilts were cleaned and moved them to the farrowing rooms on day 110 of pregnancy. The backfat thickness of gilts at the P2 position was assessed at day 110 of gestation using A-mode ultrasonography (Renco Lean-Meater^®^, Minneapolis, MN, USA). The sow’s weight was measured at days 1 and 110 of gestation. The number and the birth weights of newborn piglets were documented. To reduce the differences between placental samples, the placenta located at 3 to 4 cm from the cord insertion point was immediately collected based on previous study [18]. The samples were immediately snap-frozen in liquid nitrogen, or fixed in 4% paraformaldehyde as described previously [19].

### 2.2. Data Collection and Sampling

Experiment 1 and 2: Gilts in each group were selected for blood sampling. A total of 5 mL of their blood were collected from the auricular vein, and centrifuged at 3000× *g* and 4 °C for 15 min to recover the serum.

Experiment 2: On postnatal day 28, piglets (*n* = 6/group) with body weight close to mean weaning weight in the CON and 0.15% Orn groups were slaughtered, respectively. About 2 g jejunum and ileum samples were collected and immediately snap-frozen in liquid nitrogen. The colon contents were collected and stored at −80 °C for further analysis.

### 2.3. RNA Extraction and Real-Time Quantitative RT-PCR

Total RNA was extracted from the placenta and intestine tissues using the TRIzol reagent (Sangon Biotech, Shanghai, China). cDNA synthesis was performed according to the instructions of the PrimeScript TM regent kit (TaKaRa, Beijing, China). Primers are listed in Table 2. The real-time quantitative PCR reaction system was conducted and relative gene expression level was calculated as described previously [20].

### 2.4. Hematoxylin-Eosin (H&E) and Immunofluorescence Staining

Intestinal tissues fixed in 4% paraformaldehyde were incorporated into paraffin. Sections with a thickness of 5 μm were cut and stained with H&E. Five images of each slide were randomly selected for analysis using ImageJ software 1.48v (National Institutes of Health, Bethesda, MD, USA). Placental tissues fixed in 4% paraformaldehyde were paraffin-embedded and sectioned at 5 μm thickness for platelet endothelial cell adhesion molecule-1 (CD31) immunofluorescence as described previously [21]. Five photographs of each slide were selected randomly using a fluorescent microscope (Leica DMi8, Wetzlar, Germany). The fluorescence intensity of platelet endothelial cell adhesion molecule-1 (CD31) was quantified using Image J software 1.48v.

### 2.5. Cell Culture

Porcine vascular endothelial cells (PVECs) were purchased from the Cell Bank of the Chinese Academy of Sciences (Shanghai, China) and cultured in 1640 medium supplemented with 10% FBS, 100 U/mL penicillin, and 100 μg/mL streptomycin at 37 °C in 5% CO_2_ atmosphere.

### 2.6. ELESA Assay

The VEGF-A concentration in the culture supernatants was detected through an ELESA kit (Cusabio, Wuhan, China) according to the manufacturer’s protocol.

### 2.7. Analysis of Polyamine Content in Placenta

Approximately 25 mg placental tissue was mixed with 5 mL of 4 M perchloric acid solution at −20 °C and overnight for homogenization. The mixture was subjected to centrifugation at 3000× *g* and 4 °C for 10 min. After centrifugation, the supernatant was collected and filtered through a 0.45 μm membrane before analysis. The putrescine contents in the supernatant were subsequently determined using high performance liquid chromatography following the previously described method [8].

### 2.8. Analysis of Free Amino Acid Content in Placenta

Approximately 0.2 g of placenta sample was hydrolyzed in 10 mL of a 0.1 mol/L hydrochloric acid solution for 30 min. The mixture was subjected to centrifugation at 10,000× *g* and 4 °C for 10 min. After centrifugation, 1 mL of the supernatant was mixed with 1 mL of 8% 5-sulfosalicylic acid and then centrifuged at 12,000× *g* for 15 min. The supernatant was filtered through a 0.45 μm membrane before analysis. The Orn, arginine, proline, and glutamate contents in the supernatant were subsequently determined using high performance liquid chromatography, following the previously described method [22].

### 2.9. Small Interfering RNA (siRNA) Transfection

PVECs were plated in 6-well plates at a density of 2 × 10^4^ cells/well. After incubated for 12 h, PVECs were transfected with 20 nM siNC or siVEGF-A for 6 h. After transfection, the medium was replaced with RIPM1640 containing 10% FBS and incubated for an additional 48 h. The sequences were 5′-CCACUGAGGAGUUCAACAUTT-3′ and 5′-AUGUUGAACUCCUCAGUGGTT-3′ for VEGF-A siRNA; 5′-UUCUCCGAACGUGUCACGUTT-3′ and 5′-ACGUGACACGUUCGGAGAATT-3′ for negative control siRNA. The siRNA was purchased from Hanbio Biotechnology Co., Ltd. (Wuhan, China).

### 2.10. Tube Formation Assay

PVECs were seeded in 96-well plates precoated with 50 μL Matrigel (BD company, Franklin Lakes, NJ, USA) at a density of 4 × 10^4^ cells per well. After 6 h of treatment, images were captured using a microscope (Leica DMi8, Germany). Tube formation of PVECs was analyzed using Image J software 1.48v.

### 2.11. Transwell Assay

PVECs were seeded in the upper chamber at a density of 4 × 10^4^ cells per well with 600 μL of serum-free medium in the lower chamber. Following a 48-h incubation, cells were stained with crystal violet solution. Images were captured using a microscope (Leica DMi8, Germany) and analyzed using the Image J software 1.48v.

### 2.12. mRNA Stability Assay

PVECs were treated with 10 μg/mL actinomycin D (MedChemExpress, Monmouth Junction, NJ, USA). After incubation for indicated time points, RNA samples were extracted using TRIzol reagent (Invitrogen, Carlsbad CA, USA). The mRNA level of VEGF-A was detected by RT-PCR.

### 2.13. Analyses of Microbial Communities

The DNA of colonic microbial communities was extracted using the HiPure Stool DNA kit B (Magen, Shanghai, China). The V3 to V4 variable region of the 16S rRNA genes was amplified using the primers 515F (5′-GTGCCAGCMGCCGCGGTAA-3′) and 806R (5′-GGACTACHVGGGTWTCTAAT-3′). The library preparation and sequencing were performed by the Illumina HiSeq platform (Majorbio, Shanghai, China). Microbial analyses were performed following previously described methods [23].

### 2.14. Determination of Short Chain Fatty Acids (SCFAs) in Colonic Contents

The concentrations of acetate, propionate, butyrate, isobutyrate, valerate, and isovalerate in colonic contents were analyzed as described previously [24]. The fresh colonic content was dissolved in saline, followed by centrifugation at 10,000 rpm for 10 min to recover the supernatant fluid. One milliliter of 5% metaphosphoric acid solution was added to the supernatant fluid and vortex to mix. The mixture was centrifuged at 12,000 rpm for 10 min, then stored at 4 °C for 2 h. The supernatant was filtered through a 0.45 μm polysulfone membrane and analyzed using the Agilent 6890 gas chromatograph (Agilent Technologies, Inc., Palo Alto, CA, USA).

### 2.15. Statistical Analysis

Data are represented as mean ± SEM. Statistical analyses were performed using SPSS 20.0 (SPPS Inc., Chicago, IL, USA) software. The student’s *t*-test was performed to analyze differences between the two groups, with *p* < 0.05 considered as statistically significant.

## 3. Results

### 3.1. Fetal Weight and Amino Acids Contents in Placenta

To investigate the placental angiogenesis of low-weight fetuses during pregnancy, four gilts were euthanized at days 75 (*n* = 4) and 110 (*n* = 4) of gestation, respectively. We found that the body weight of fetal pigs at day 110 of gestation was higher (*p* < 0.05) than that at day 75 of gestation (Figure 1A). The contents of Orn, proline, and putrescine in the placenta at day 75 of gestation were higher (*p* < 0.05) than those at day 110 of gestation, whereas no significant difference (*p* > 0.05) was detected in the levels of arginine and glutamate between these two-time points (Figure 1B–F). Subsequently, the placentae of fetal pigs were divided into normal weight group (N) and low weight group (L) based on fetal pig’s body weight (Figure 1G). The levels of the placental Orn, arginine, and putrescine were significantly lower in L group (*p* < 0.05) than those in N group at day 75 of gestation. (Figure 1H,I,L). No significant change (*p* > 0.05) was observed in the contents of proline and glutamate between the two groups (Figure 1J,K).

### 3.2. Placental Vascular Density

Placental vessels are vital for growth and development of fetal pigs [25]. Therefore, the placental angiogenesis of low weight fetuses at days 75 and 110 of gestation were determined. The results indicated that the L group was lower (*p* < 0.05) than the N group in the mRNA expression levels of CD31 and VEGF-A in placenta (Figure 2A,B). Compared with the N group, placentae in L group showed a decreased (*p* < 0.05) fluorescence intensity of CD31 at days 75 and 110 of gestation (Figure 2C–F).

### 3.3. Reproductive Performance of Sows

To investigate the role of Orn in placental angiogenesis, gilts were fed diets supplemented with 0.05% Orn, 0.10% Orn, 0.15% Orn. Results showed that no differences (*p* > 0.05) were observed in the maternal body weight and backfat thickness at day 110 of gestation among the four groups (Table 3). The serum Orn content and reproduction performance of sows were shown in Table 4. The serum Orn concentration was higher (*p* < 0.05) in the 0.10% Orn and 0.15% Orn groups than that in the CON group. The litter size or born alive was not affected (*p* > 0.05) by diet Orn supplementation. Compared with the CON group, maternal dietary supplementation with 0.10% or 0.15% Orn increased (*p* < 0.05) the birth weight of piglets. No difference (*p* > 0.05) was observed in the ADFI of sows among four groups during lactation. The body weight and average daily gain (ADG) of suckling piglets at postnatal day 28 was not affected (*p* > 0.05) by diet Orn supplementation. These findings indicate that maternal supplementation with 0.15% Orn increased the birth weight of piglets.

### 3.4. Maternal Orn Supplementation Promotes Angiogenesis in Placental Vascular Density

The results showed that the mRNA level and immunostaining intensity of CD31 were significantly increased (*p* < 0.05) in the 0.15% Orn group relative to the CON group (Figure 3A–C). The VEGF-A level in the placenta was elevated (*p* < 0.05) in the 0.15% Orn group than that in the CON group (Figure 3D). Moreover, the mRNA expression levels of amino acid transport-related factors in the placentae were detected using real-time PCR. The results showed that the mRNA expression levels of ASC amino acid transporter-2 (ASCT2), proton-coupled amino acid transporter 2 (PAT2), and sodium-coupled neutral amino acid transporter 2 (SNAT2) were significantly higher (*p* < 0.05) in the 0.15% Orn group than those in the CON group (Figure 3E). The above-described results illustrate that maternal supplementation with 0.15% Orn promotes angiogenesis in placenta.

### 3.5. VEGF-A Mediates Angiogenesis of Orn Controlled

To further investigated the potential mechanism of Orn on placental angiogenesis, PVECs were treated with Orn. The results showed that the PVECs viability was significantly augmented (*p* < 0.05) after being treated with 200 μM Orn for 48 or 72 h contrasted with the CON group (Figure 4A). A 200 μM Orn treatment was found to promote (*p* < 0.05) PVECs tube formation and migration (Figure 4B–D). Subsequently, the mRNA levels of genes related to angiogenesis were detected. The results showed that the mRNA and protein levels of VEGF-A in PVECs were elevated (*p* < 0.05) in the Orn group than that in the CON group (Figure 4E,F). Furthermore, a 200 μM Orn treatment increased (*p* < 0.05) the mRNA stability of VEGF-A in PVECs (Figure 4G). To examine the hypothesis that Orn promotes angiogenesis by enhancing the expression level of VEGF-A, siRNA was utilized to reduce the expression level of VEGF-A (Figure 4H). After knockdown of VEGF-A, no significant difference (*p* > 0.05) was identified in proliferation and tube formation between si NC and si VEGF-A + Orn group (Figure 4I–K). Overall, these results suggest that Orn enhances the placental angiogenesis via VEGF-A.

### 3.6. Intestinal Development of Suckling Piglets

Maternal nutrition exerts a vital role in intestinal development of piglets. Therefore, we investigated the influence of Orn on intestinal development of sucking piglets. In comparison to the CON group, the 0.15% Orn group was higher (*p* < 0.05) than the CON group in jejunal and ileal villi height (Figure 5A,B,E). No difference (*p* > 0.05) was observed in crypt depth and the ratio of V/C in jejunum (Figure 5C,D). Compared with CON group, the ileal crypt depth was decreased (*p* < 0.05) and V/C was increased (*p* < 0.05) in 0.15% Orn group (Figure 5F,G). Furthermore, a real-time PCR was performed to determine the differences in the mRNA expression levels of intestinal barrier-associated factors in the jejunum and ileum. The results showed that 0.15% Orn group was higher (*p* < 0.05) than the CON group in the mRNA expression levels of *ki67*, *occludin*, *Mucin 1* (*MUC1)*, and *Mucin 2* (*MUC2)* in the jejunum and ileum (Figure 5H,I). The above-described results illustrate that maternal supplementation with 0.15% Orn promotes the intestinal morphology of suckling piglets.

### 3.7. Colonic Bacterial Community Structure

Next, we explored the effects of Orn on the colonic bacterial community in sucking piglets. The results showed that the most dominant phyla among colonic bacterial communities were *Firmicutes*, *Bacteroidetes*, *Spirochaetae*, and *Tenericutes* (Figure 6A), collectively representing over 90% of the total bacterial composition found in the colonic content. The abundance of *Tenericutes* was significantly higher (*p* < 0.05) in the 0.15% Orn group than that in the CON group (Figure 6C). In addition, a decreasing trend (*p* = 0.06) was identified in the abundance of *Spirochaetae* in the 0.15% Orn group relative to the CON group (Figure 6C). Figure 6B shows the abundance distribution of the dominant bacterial genera between CON and 0.15% Orn groups. The abundances of *Clostridium_sensu_stricto_1*, *Ruminococcaceae_UCG-014*, *Mollicutes_RF9*, and *Erysipelotrichaceae* were higher, while that of *Bacteroidales_BS11_gut_group* were lower in the 0.15% Orn group than those in the CON group (Figure 6D). Collectively, these findings indicate that maternal supplementation with 0.15% Orn alters the colonic bacterial community structure of sucking piglets.

### 3.8. SCFAs in Colonic Contents of Suckling Piglets

Short chain fatty acids (SCFAs) play important role in maintaining the intestine homeostasis [26]. As shown in Figure 7, no significant difference (*p* > 0.05) was identified in the concentrations of acetate, valerate, isobutyrate, or isovalerate between CON and 0.15% Orn group (Figure 7A,D–F). However, the 0.15% Orn group was higher (*p* < 0.05) than the CON group in the concentrations of propionate and butyrate (Figure 7B,C). Overall, these findings demonstrate that maternal supplementation with 0.15% Orn increased the contents of propionate and butyrate in colonic contents.

## 4. Discussion

Placental blood vessels play an integral role in the exchange of gases, nutrients, and waste materials between mother and fetus, which is crucial for fetal growth and development [27,28]. In the present study, the density of placental blood vessels was found to be significantly lower in low-weight piglets than that in normal-weight piglets, aligning with previous study [29]. Furthermore, we observed a decrease in the contents of Orn and putrescine in the placentae of low-weight piglets at day 75 of gestation. Orn is known to play a vital role in angiogenesis [30,31]. Additionally, products of Orn, including putrescine, spermidine, and spermine, were reported to facilitate placental angiogenesis [32]. Therefore, we hypothesized that an Orn deficiency in low-weight piglet placentae could attenuate the placental angiogenesis. To explore this hypothesis, Orn was added to the diet of pregnant gilts. As expected, maternal dietary supplementation with 0.15% Orn led to an increase in piglet birth weight and increased the angiogenesis of placenta. Orn was found to reduce H_2_O_2_-induced ROS production [33]. Increased oxidative stress could decrease angiogenesis via causing vascular endothelial cell injury [34]. Therefore, Orn may enhance placental angiogenesis by reducing the levels of oxidative stress in placenta. In addition, Orn can be converted to citrulline and glutamate [35]. Previous studies indicated that glutamate can activate the mTORC1 pathway by generating aspartate, thereby promoting angiogenesis [36]; citrulline could improve placental angiogenesis in gilts [37]. Orn is also the substrate for the synthesis of polyamines. Evidence confirmed that spermidine improves angiogenesis of senescent endothelial cells and induces neovascularization; putrescine promotes angiogenesis via hydrogen peroxide/METTL3 pathway [38,39]. The above-mentioned results indicate that Orn promote placental angiogenesis might by inducing citrulline, glutamate, and polyamines generation. A previous study showed that the expression levels of amino acid transporters were significantly lower in early preterm IUGR complicated placentae [40]. Consistent with this result, the mRNA expression levels of *ASCT*2, *PAT*2, and *SNAT*2 were upregulated with 0.15% Orn supplementation. Collectively, these findings suggest that maternal Orn supplementation enhances the placental angiogenesis and the birth weight of piglets. Pigs are commonly used as a model to investigate human diseases [41]. These results also contribute to improve our understanding of the application of Orn in promoting fetal growth and placental angiogenesis.

Placental angiogenesis is governed by angiogenic factors [42]. In this study, the levels of mRNA and protein of VEGF-A in placenta were increased under maternal Orn treatment. VEGF-A is one of the critical angiogenic inducers that is known to promote placental angiogenesis via stimulating endothelial cell proliferation and migration [29]. To investigate whether Orn modulates angiogenesis through VEGF-A, we established stable VEGF-A knockdown in PEVCs. As expected, knockdown of VEGF-A in PEVCs blocked the promoting effect of Orn on PEVCs angiogenesis in vitro. Mechanistically, Orn augmented VEGF-A expression by enhancing the mRNA stability of VEGF-A. N6-methyladenosine (m6A) modification, one of the most common modifications of mRNAs [43], was previously shown to boost the translation of VEGF-A to accelerate angiogenesis [44], suggesting m6A modification might be involved in the process of Orn in modulating VEGF-A mRNA stability, though this mechanism warrants further investigation. The above-described indicated that Orn promotes placental angiogenesis through regulating VEGF-A.

Maternal nutrition during gestation is an important factor influencing the intestinal development of postnatal piglets. Numerous studies have elucidated that maternal dietary supplementation with specific nutrients, such as vitamin D3, galactooligosaccharides, 25-hydroxycholecalciferol, can improve the intestinal development and health of suckling piglets [45,46,47]. In the present investigation, the 0.15% Orn group was higher than the CON group in the jejunal and ileal villi height. These findings are in agreement with previous research in mice, demonstrating that administration of lactobacilli promote gut mucosal formation through Orn production [48]. The proliferation of intestinal epithelial cells is vital for maintaining intestinal epithelial function and renewal [49]. Earlier studies showed that increased cell apoptosis and decreased proliferation of intestinal epithelial cells coincide with a reduction in villi height [50,51]. Consistently, the mRNA level of ki67 in jejunum and ileum of suckling piglets was higher in 0.15% Orn group, suggesting maternal supplementation with Orn during gestation increased villi height might via fostering the proliferation of intestinal epithelial cells. The small intestine is not only the main site for nutrients digestion and absorption but also constitutes an essential defense line against the invasion of bacteria and endotoxins into the intestinal lumen [52,53]. Proteins such as Occludin, a tight junction protein, are essential for maintaining intestinal structural integrity [54]. MUC1 and MUC2, belongs to mucins family, form components of the mucus gel layer, thereby protecting intestinal epithelial cells from extreme environmental conditions [55]. Here, Orn treatment increased the mRNA expression levels of Occludin, MUC1, and MUC2 in jejunum and ileum of suckling piglets, suggesting maternal Orn supplementation with Orn improved the intestinal barrier of suckling piglets.

Microbiota plays an essential role in maintaining intestinal mucosal integrity and barrier [56]. For instance, microbiota sensing by Mincle-Syk axis in dendritic cells promotes intestinal barrier integrity [57]. Consequently, we examined the composition of the gut microbiota in the present study. The results showed that a decreasing trend was found in the abundance of *Spirochaetae* by maternal Orn supplementation. *Spirochaetaceae*, a pathogenicbacteria, has the potential to compromise intestinal epithelial cells of animals [58], suggesting that maternal Orn supplementation improves the intestine development though reducing the abundance of *Spirochaetae*. Furthermore, the abundances of Clostridium_sensu_stricto_1 and Mollicutes_RF9 were increased with Orn treatment. Clostridium_sensu_stricto_1 could promote intestinal development by producing SCFAs [59,60], while increased Mollicutes_RF9 abundance has been associated with augmented butyrate production [61]. The microbial metabolites of SCFAs play crucial role in maintaining gut and metabolic health [62]. Previous studies showed that propionate regulates tight junction barrier by increasing endothelial-cell selective adhesion molecule [63], while butyrate enhances the intestinal barrier by facilitating tight junction assembly via activating AMP-activated protein kinase [64]. Concomitant with the changes in the abundances of Clostridium_sensu_stricto_1 and Mollicutes_RF9, we found that maternal Orn supplementation during gestation significantly increased the levels of propionate and butyrate in colonic contents of suckling piglets. Collectively, these above results suggested that maternal Orn supplementation during gestation facilitates the intestine development and health of the offspring through modifying the composition of microbiota and engendering propionate and butyrate.

## 5. Conclusions

In conclusion, the study showed that the vascular density and Orn level were significantly decreased in placenta of low-birth-weight piglets. Maternal dietary supplementation with 0.15% Orn was found to stimulate placental angiogenesis by targeting VEGF-A. Moreover, 0.15% Orn supplementation promotes intestine development of sucking piglets. In conclusion, these findings suggest that maternal Orn supplementation represents an efficacious nutritional approach to improve fetal growth by the augmentation of placental angiogenesis and provide an important avenue for improving the intestinal development of suckling piglets.

## Figures and Tables

**Figure 1 animals-14-00689-f001:**
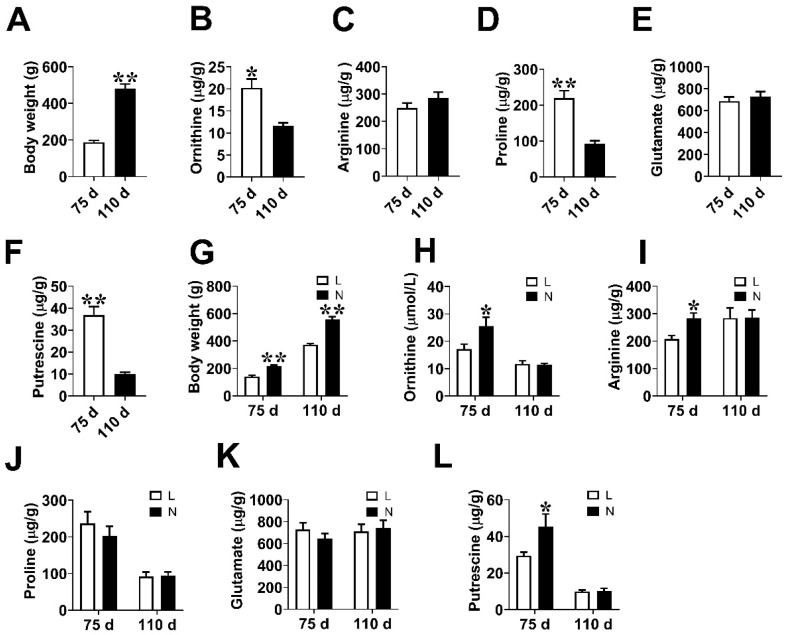
Fetal weight and amino acid content in placenta. (**A**) The body weight of fetal piglets at days 75 and 110 of gestation. (**B**–**F**) The contents of Orn, arginine, proline, glutamate, and putrescine in the placenta at days 75 and 110 of gestation, respectively. (**G**) The body weight of fetal piglets for the L and N groups at days 75 and 110 of gestation. (**H**–**L**) The contents of Orn, arginine, proline, glutamate, and putrescine in the placenta at days 75 and 110 of gestation, respectively. *n* = 12. Data are presented as mean ± SEM. * indicates significant differences at *p* < 0.05, ** indicates significant differences at *p* < 0.01.

**Figure 2 animals-14-00689-f002:**
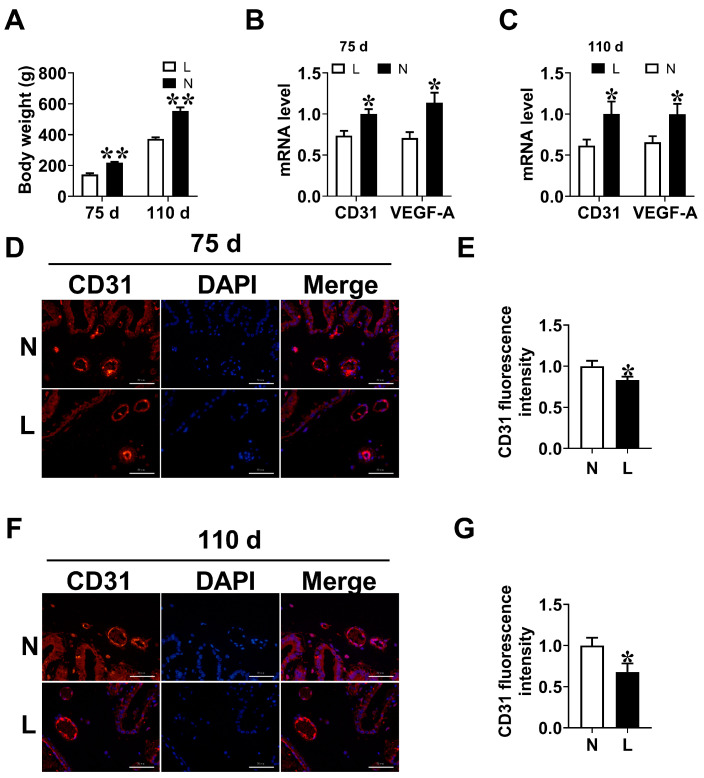
The vascular density of placenta. (**A**) The body weight of fetal piglets at days 75 and 110 of gestation. (**B**,**C**) The mRNA expression levels of *CD31* and *VEGF-A* in the placenta at days 75 and 110 of gestation. (**D**–**G**) CD31 immunofluorescence staining in the placenta at days 75 and 110 of gestation. bar = 50 μm.Data are presented as mean ± SEM. *n* = 12. * indicates significant differences at *p* < 0.05, ** indicates significant differences at *p* < 0.01.

**Figure 3 animals-14-00689-f003:**
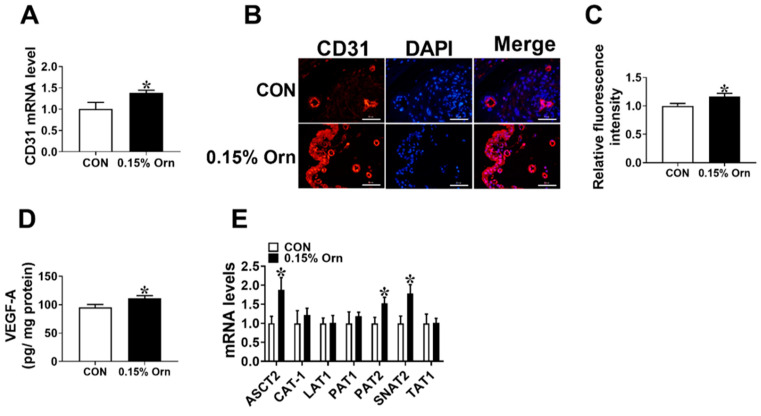
Effect of maternal Orn supplementation during gestation on placental anagenesis. (**A**) The mRNA expression level of *CD31* in the placenta. bar = 50 μm. (**B**,**C**) CD31 immunofluorescence staining of the placenta. (**D**) The content of VEGF-A in the placenta. (**E**) The mRNA expression levels of genes related to amino acid transporters. Data are presented as mean ± SEM. *n* = 12. * indicates significant differences at *p* < 0.05.

**Figure 4 animals-14-00689-f004:**
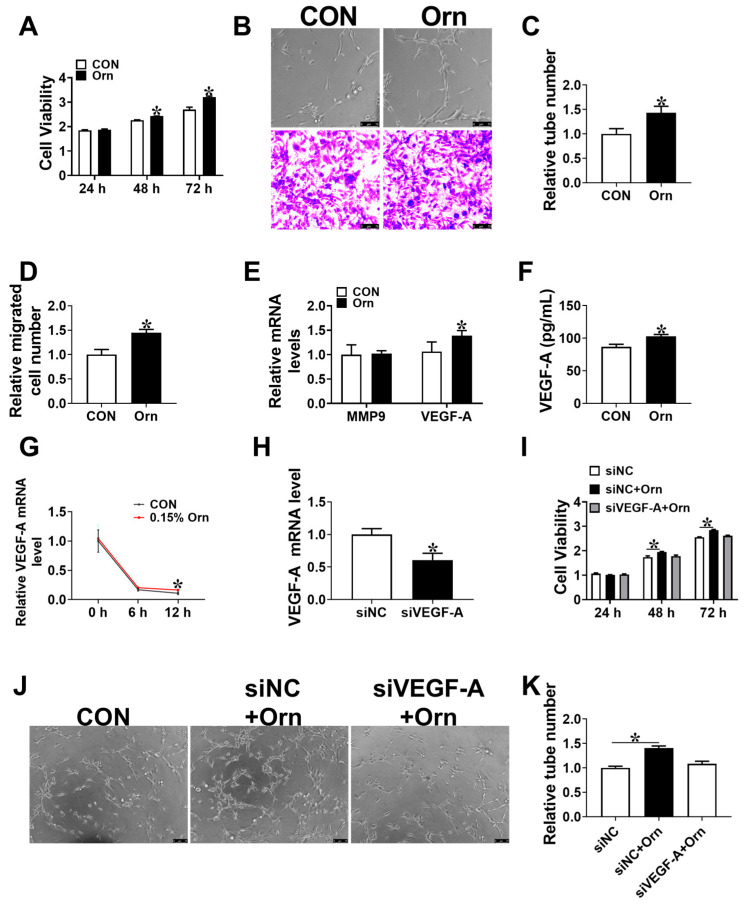
Ornithine stimulates angiogenesis in vitro via VEGF-A. (**A**) Proliferation assay of PVECs treated with 200 μM Orn for intervals of 24, 48, and 72 h (*n* = 6). (**B**–**D**) PVECs tube formation (**upper**) and migration (**below**) (*n* = 5). bar = 75 μm. (**E**) The mRNA expression levels of MMP9 and VEGF-A in PVECs treated with 200 μM Orn for 48 h (*n* = 6). (**F**) The concentration of VEGF-A in cultured media (*n* = 6). (**G**) The mRNA stability of VEGF-A, as examined in PVECs treated with 200 μM Orn for 48 h, subsequently treated with 10 μg/mL actinomycin D for intervals of 0, 6, and 12 h. VEGF-A mRNA level was analyzed by quantitative PCR (qPCR) (*n* = 6). (**H**) qPCR evaluation of VEGF-A mRNA expression level in PVECs with siRNA against VEGF-A for 48 h (*n* = 6). (**I**) Proliferation assay of PVECs (*n* = 6). (**J**,**K**) The representative micrographs of tube formation in PVECs (*n* = 6). bar = 75 μm. Data are depicted as mean ± SEM, and * indicates significant differences at *p* < 0.05.

**Figure 5 animals-14-00689-f005:**
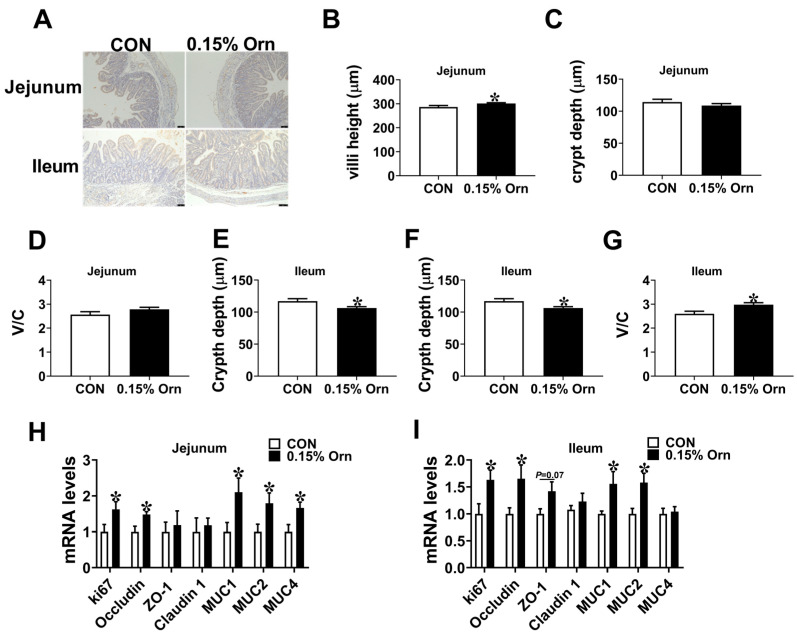
Influence of maternal Orn supplementation during gestation on intestinal morphology of suckling piglets. (**A**) Hematoxylin and eosin (H&E) staining of jejunum and ileum. bar = 75 μm. (**B**–**G**) Analysis of the villi height, crypt depth, and the ratio of villi height to crypt depth in both jejunum and ileum. (**H**,**I**) The mRNA expression levels of factors related to intestinal barriers in jejunum and ileum. Data are expressed as mean ± SEM, *n* = 12, with * indicates significant differences at *p* < 0.05.

**Figure 6 animals-14-00689-f006:**
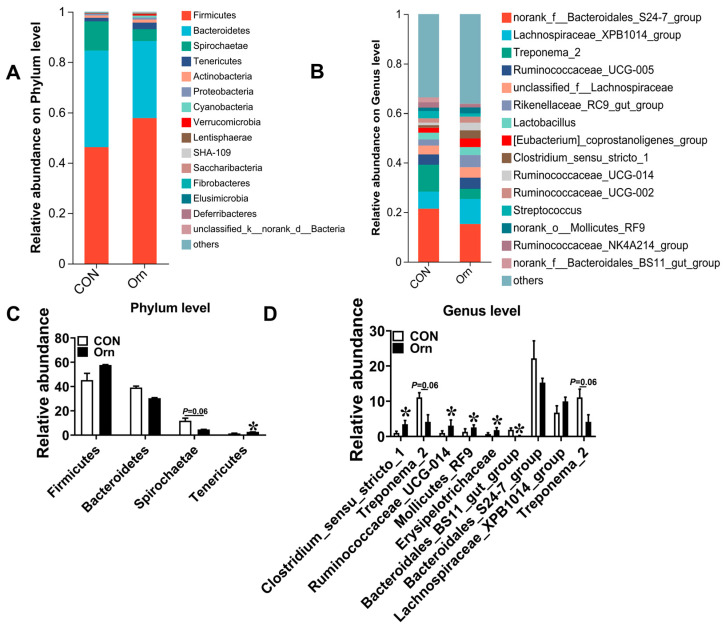
Impact of maternal Orn supplementation during gestation on the colonic bacterial community of suckling piglets. (**A**) Distribution of colonic bacteria at the phylum level in suckling piglets, with phyla having proportions less than 0.01 grouped as others. (**B**) Enumeration of the top 15 abundant bacterial genera. (**C**,**D**) Disparities of colonic microbiota at the phylum and genus levels, analyzed using the Kruskal-Wallis test. Data are presented as mean ± SEM, *n* = 6, with * signifying significant disparities at *p* < 0.05.

**Figure 7 animals-14-00689-f007:**
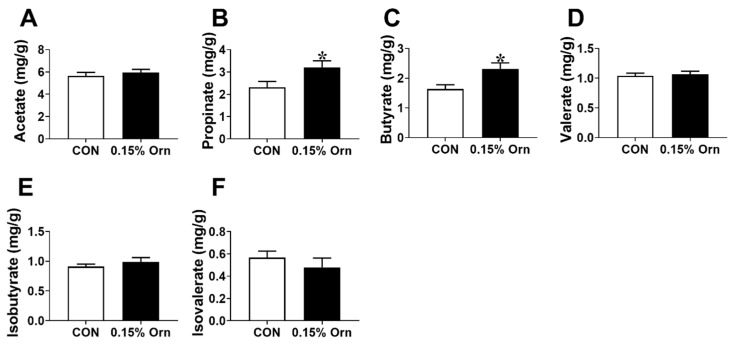
Concentration of short-chain fatty acids in the colonic contents of suckling piglets. The content of acetate (**A**), propionate (**B**), butyrate (**C**), valerate (**D**), isobutyrate (**E**), and isovalerate (**F**) in the colonic contents of sucking piglets. Data are furnished as mean ± SEM, *n* = 12, and * indicates significant differences at *p* < 0.05.

**Table 1 animals-14-00689-t001:** Composition and nutrient levels of the basal diets for gilts (as-fed basis) ^1.^

Items	Gestating Diet	Lactating Diet
Ingredient, %		
Corn	46.00	52.20
Soybean meal	13.00	20.00
Barley	12.00	10.00
Wheat bran	15.00	10.00
soybean hull	10.00	-
soybean oil	-	3.80
Premix ^1^	4	4
Total	100.00	100.00
Chemical composition		
DE, MJ/kg	12.33	14.00
CP, %	13.71	15.19
EE, %	2.86	6.48
ASH	2.75	2.63
CF, %	6.80	3.32
Ca, %	0.85	0.82
P, %	0.61	0.60
Lys, %	0.66	0.76
Met + Cys, %	0.54	0.55
Trp, %	0.16	0.18
Thr, %	0.53	0.60

^1^ Gestating gilts’ premix provided the following per kg of the diet: Cu, 25 mg; Se, 10 mg; Fe,120 mg; Mn, 35 mg; Zn, 75 mg; vitamin A, 12,000 IU; vitamin B_2_, 80 mg; vitamin D_3_, 1500 IU; vitamin E, 50 IU; vitamin K_3_, 50 mg; vitamin B_2_, 10 mg; vitamin B_6,_ 15 mg; calcium pantothenate, 150 mg; vitamin B_12_, 0.1 mg, limestone, 12 g. Lactating gilts’ premix provided the following per kg of the diet: Cu, 28 mg; Se, 12 mg; Fe,150 mg; Mn, 40 mg; Zn, 75 mg; vitamin A, 13,000 IU; vitamin B_2_, 100 mg; vitamin D_3_, 1500 IU; vitamin E, 70 IU; vitamin K_3_, 70 mg; vitamin B_2_, 15 mg; vitamin B_6,_ 10 mg; calcium pantothenate, 150 mg; and vitamin B_12_, 0.2 mg.

**Table 2 animals-14-00689-t002:** Primers used for real-time PCR.

Genes	Primers	Sequences (5′ to 3′)
ASCT2	Forward	GATTGTGGAGATGGAGGATGTGG
	Reverse	TGCGAGTGAAGAGGAAGTAGATGA
CAT1	Forward	TGCCCATACTTCCCGTCC
	Reverse	GGTCCAGGTTACCGTCAG
CD31	Forward	TGTGGACTTCTTCCTGATTGTC
	Reverse	CGTTGTTGTGGCAGTTGTTGGT
Claudin-1	Forward	GATTTACTCCTACGCTGGTGAC
	Reverse	CACAAAGATGGCTATTAGTCCC
Occludin	Forward	GCACCCAGCAACGACAT
	Reverse	CATAGACAGAATCCGAATCAC
PAT1	Forward	TGTGGACTTCTTCCTGATTGTC
	Reverse	CGTTGTTGTGGCAGTTGTTGGT
PAT2	Forward	GGGCTACTTGCGGTTCGG
	Reverse	GCGCTTTGACACCTGGGAG
TAT1	Forward	GCCCATTGCCTTCGAGTTAG
	Reverse	AGCGAGGTAGAATGCCACAT
LAT1	Forward	TTTGTTATGCGGAACTGG
	Reverse	AAAGGTGATGGCAATGAC
SNAT2	Forward	TACTTGGTTCTGCTGGTGTCC
	Reverse	GTTGTGGGCTGTGTAAAGGTG
VEGF-A	Forward	CCTCGGAGCGGAGAAAGCAT

**Table 3 animals-14-00689-t003:** Effects of dietary supplementation with ornithine during gestation on body condition of sows (*n* = 12/group).

Items	CON	0.05% Orn	0.10% Orn	0.15% Orn	*p*-Value
Body weight of sows, kg					
Day 1	91.71 ± 6.11	91.88 ± 3.23	89.79 ± 4.91	91.29 ± 4.89	0.99
Day 110	124.21 ± 4.19	124.42 ± 3.41	125.37 ± 4.59	126.92 ± 4.75	0.97
Body weight gain	32.50 ± 3.17	33.54 ± 2.76	35.58 ± 1.41	35.63 ± 2.02	0.66
Backfat thickness of sows, mm					
Day 1	29.25 ± 0.93	29.58 ± 0.73	29.41 ± 1.12	29.08 ± 0.71	0.98
Day 110	32.42 ± 0.83	32.58 ± 0.50	32.83 ± 0.79	32.25 ± 0.48	0.93
Backfat thickness gain	3.16 ± 0.42	2.92 ± 0.46	3.42 ± 0.71	3.16 ± 0.34	0.92

**Table 4 animals-14-00689-t004:** Effects of dietary supplementation with ornithine during gestation on serum Orn content, reproduction, and lactation performance of sows (*n* = 12/group).

Items	CON	0.05% Orn	0.10% Orn	0.15% Orn	*p*-Value
Serum Orn, μmol/L	55.81 ± 3.24 ^b^	58.53 ± 2.09 ^b^	66.28 ± 1.87 ^a^	68.65 ± 1.71 ^a^	<0.01
litter size	10.50 ± 0.73	10.42 ± 0.45	10.75 ± 0.78	10.75 ± 0.68	0.98
Born alive	9.33 ± 0.69	9.83 ± 0.42	9.58 ± 0.70	9.75 ± 0.85	0.96
Stillborn piglets	1.17 ± 0.24	0.90 ± 0.25	1.17 ± 0.34	1.18 ± 0.50	0.95
Birth weight, g	590.92 ± 17.25 ^b^	601.42 ± 19.55 ^b^	654.27 ± 15.80 ^a^	665.27 ± 19.98 ^a^	0.01
Lactation					
ADFI of sows, kg	2.61 ± 0.24	2.56 ± 0.25	2.59 ± 0.39	2.55 ± 0.38	0.89
Piglet weight at weaning, g	3423.29 ± 120.89	3291.07 ± 87.15	3406.95 ± 128.53	3529.25 ± 69.80	0.47
ADG of piglets, g/d	100.61 ± 4.10	96.78 ± 3.68	97.40 ± 4.81	102.38 ± 2.97	0.71

ADG: Average daily gain; ADFI: Average daily feed intake. ^a–b^ Values are means with pooled SEM. Within a row, means not sharing the same superscript letters differ significantly.

## Data Availability

The original contributions generated for this study are included in the article, further inquiries can be directed to the corresponding author.

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
