# Peer review of "Maternal Supplementation with Ornithine Promotes Placental Angiogenesis and Improves Intestinal Development of Suckling Piglets"

_animals, 2024, doi:10.3390/ani14050689_

Round 1
Reviewer 1 Report
Comments and Suggestions for Authors
Lines 25 and 79. Says: 10 % orn (0.05 % orn group). Should say (0.10 % orn group).
Lines 63-74: It is not clear to the reader how the 8 mentioned sows were used in the experiment after being slaughtered.
Table 4: I consider that the percentages of stillborn piglets should also be mentioned, and whether these died before or after birth. The mummies should also be mentioned, and if there could be any relationship between these parameters and the treatments.
In Figure 5 G, I and K, comparisons are being made between three groups or times. Why student’s t test was used?
Reviewer 2 Report
Comments and Suggestions for Authors
General Comments:
Results are reported on pig production traits, specific genes of interest responses in placental and intestinal tissues, and microbial profiles of pigs from sows fed control or ornithine supplemented diets. The inferences are that Orn supplements increase placenta tissue angiogenesis, intestinal integrity and alters colonic microbiota, which collectively improves neonatal pig growth. The conclusion that this response is specific to Orn cannot be supported by the experimental design. Would supplements of citrulline, glutamate, or aspartate give the same response in the diets imposed. Are the responses simply mediated by alterations of ammonia and urea metabolism? The authors must consider the complex integration of amino acid metabolism rather than conclude responses are specific to only one non-essential amino acid.
Specific Comments:
L 41 fetuse vs fetus
L 72 - 73. Suggest "Fetal and placenta weights were recorded. Two grams of each placenta was immediately......"
L 82. What percentage of the daily energy requirement is met by 2.5 kg feed. This seems to be quite restricted.
Table 1. Should Wheat brain be Wheat bran?
Are the Ca concentrations correct? If these values represent concentrations in the complete diet, I would expect major health issues with the sows. Apparently, no inorganic Ca or P ingredients were supplemented.
Please double check the concentrations for vitamin D, 15,000 IU.kg diet seems excessive and if the vitamin A concentrations are correct concerns would exist for a vitamin D X vitamin A interaction.
L 98. In L 73 you stated that 2 g of tissue was collected?
L 99. Was the placenta tissue location collected in a manner to allow specific placenta location to be associated with the individual fetal pig? Maybe I am confusing Trial 1 and Trial 2. If so, can you re-word so future readers are not also confused.
L 136. Should this be ELISA ?
L 260 to 270. The logic for cell culture assays is not clearly explained. Would the same responses be detected if glutamate, aspartate, or citrulline was used as a treatment?
L 397 to 402. The results support, in part, the statement "maternal Orn supplementation during gestation significantly increased the levels of propionate and butyrate in colonic contents of suckling piglets". However, the experimental design precludes a conclusion that the response is specific to Orn, as other non-essential amino acids were not applied as a control. See general comments.
Reviewer 3 Report
Comments and Suggestions for Authors
This manuscript addresses a significant and specific scientific question, providing valuable insights into the role of ornithine in placental angiogenesis and the development of piglets. The findings are potentially impactful for animal nutrition and developmental biology.
Introduction: There is little written about the research background, especially why we should study the influence of maternal Orn supplementation on the intestinal development of offspring.
The introduction and discussion sections could benefit from a clearer articulation of how this study advances the current knowledge in the field.
Line 64: What are the average weight and age of these pigs?
Line 68-69: What are the types of the feeders and drinkers?
Line 71-72: How did these gilts get pregnant? What kind of boars? How many were there?
Line 64-75: These pigs had not been treated at all? Why kill them and take samples?
Line 76: What are the average weight and age of these pigs?
Line 84: How did these gilts get pregnant? What kind of boars? How many were there?
Line 96: what’s “P2 position”?
Line 95-107: You have two batches of experimental gilts. Which batch of experimental gilts did these data refer to?
Line 104: “On postnatal day 28, piglets (n =12) with similar mean birth weight were slaughtered.” How many piglets in each group? How about their gender? What was the average weight of kg?
Line 204: These should be in the Material and Methods. What were the criteria and basis for grouping?
Figure 2, 4: I can't see immunofluorescence staining clearly.
Figures and tables should be more effectively integrated into the narrative to aid reader comprehension.
Table 3: sows (n=12)?
Table 4: How many piglets were there in each group?
The paper could explore potential broader applications of the findings or implications for other species.
Some sections require a more detailed explanation of the underlying mechanisms and implications of the findings.
Round 2
Reviewer 2 Report
Comments and Suggestions for Authors
Thank you for the changes and responses to the first review
Author Response
|
3. Point-by-point response to Comments and Suggestions for Authors Thank you for the changes and responses to the first review Reply: Thank you for your comment. |
|
4. Response to Comments on the Quality of English Language |
English language fine. No issues detected
Reply: Thank you for your comment.

Reviewer 3 Report
Comments and Suggestions for Authors
The authors addressed my concerns. But there are some minor problems that need to be addressed before publication.
Line 130: change “gene expression”to “gene expression level”
Line 143: change“were purchase” to “were purchased”
Figure 2C and 2E: The image is too small to be seen clearly. Please enlarge the image.
Table 3 and Table 4: The line number reaches the left and overlaps with the table content.
Figure 4B: The image is too small to be seen clearly. Please enlarge the image.
Figure 5B and 5J: The image is too small to be seen clearly. Please enlarge the image.
Figure 6A: The image is too small to be seen clearly. Please enlarge the image.
Author Response
|
3. Point-by-point response to Comments and Suggestions for Authors |
|
The authors addressed my concerns. But there are some minor problems that need to be addressed before publication. Reply: We appreciate the reviewer’s comments. Line 130: change “gene expression”to “gene expression level” Reply: Thank you for your valuable comments. We have revised this sentence according to your comment, please see line 129. Line 143: change“were purchase” to “were purchased” Reply: Thank you for your valuable comments. We have revised this sentence according to your comment, please see line 142. Figure 2C and 2E: The image is too small to be seen clearly. Please enlarge the image. Reply: Thank you for your valuable comments. We have enlarged this image according to your comment. Please see figure 2. Table 3 and Table 4: The line number reaches the left and overlaps with the table content. Reply: Thank you for your valuable comments. We have revised the line number according to your comment. Please see table 3 and table 4. Figure 4B: The image is too small to be seen clearly. Please enlarge the image. Reply: Thank you for your valuable comments. We have enlarged this image according to your comment. Please see figure 4. Figure 5B and 5J: The image is too small to be seen clearly. Please enlarge the image. Reply: Thank you for your valuable comments. We have enlarged this image according to your comment. Please see figure 5. Figure 6A: The image is too small to be seen clearly. Please enlarge the image. Reply: Thank you for your valuable comments. We have enlarged this image according to your comment. Please see figure 6. |
|
4. Response to Comments on the Quality of English Language |
English language fine. No issues detected
Reply: Thank you for your valuable comments.
